# Toward Spatial Identities in Human Brain Organoids-on-Chip Induced by Morphogen-Soaked Beads

**DOI:** 10.3390/bioengineering7040164

**Published:** 2020-12-18

**Authors:** Lihi Ben-Reuven, Orly Reiner

**Affiliations:** Weizmann Institute of Science, Rehovot 7610001, Israel; lihi.benreuven@weizmann.ac.il

**Keywords:** organoids, brain, development, human embryonic stem cells, beads, morphogens, WNT, dorsoventral, anteroposterior, live imaging

## Abstract

Recent advances in stem-cell technologies include the differentiation of human embryonic stem cells (hESCs) into organ-like structures (organoids). These organoids exhibit remarkable self-organization that resembles key aspects of in vivo organ development. However, organoids have an unpredictable anatomy, and poorly reflect the topography of the dorsoventral, mediolateral, and anteroposterior axes. In vivo the temporal and the spatial patterning of the developing tissue is orchestrated by signaling molecules called morphogens. Here, we used morphogen-soaked beads to influence the spatial identities within hESC-derived brain organoids. The morphogen- and synthetic molecules-soaked beads were interpreted as local organizers, and key transcription factor expression levels within the organoids were affected as a function of the distance from the bead. We used an on-chip imaging device that we have developed, that allows live imaging of the developing hESC-derived organoids. This platform enabled studying the effect of changes in WNT/BMP gradients on the expression of key landmark genes in the on-chip human brain organoids. Titration of CHIR99201 (WNT agonist) and BMP4 directed the expression of telencephalon and medial pallium genes; dorsal and ventral midbrain markers; and isthmus-related genes. Overall, our protocol provides an opportunity to study phenotypes of altered regional specification and defected connectivity, which are found in neurodevelopmental diseases.

## 1. Introduction

Recent advances in stem-cell technologies include the differentiation of human embryonic stem cells (hESCs) into organ-like structures (organoids) [1,2,3] (reviewed by Di Lullo and Kriegstein, 2017 [4]). These organoids exhibit self-organization and developmental trajectories, which mimics key features of known in vivo organ development as well as gene expression patterns [5]. Different protocols can direct hESCs to generate a diversity of cell identities from neuroepithelial domains [1,3,6]. Other protocols induce regional identities, such as cortical, hippocampal, and choroid plexus tissues [7,8], or generation of forebrain, midbrain, and hindbrain organoids [9]. These characteristics make the organoids an attractive model not only to study developmental concepts, but also for disease modeling [3,4,6,9]. However, complex brain structure and function in vivo is supported by specific anterior-posterior (AP), dorso-ventral (DV), and medio-lateral positioning cues, which are difficult to achieve in cell cultures. In vivo, the induction of spatial identity is induced by the secretion of signaling molecules called morphogens from organizing centers [10]. These morphogens affect gene expression programs in a threshold- and concentration-dependent manner and their activity is regulated by the cross-talk with stimulating or opposing factors [10,11,12,13]. The main stages of neural development include formation of the neural plate, the neural groove, early and late neural tube stages (reviewed by Kiecker and Lumsden. 2012 [14]). The initial positional information across the neural plate includes mediolateral gradients of BMP and Shh, together with an anteroposterior gradient of Wnt activity. These conserved Wnt gradients mediate eventually the formation of the forebrain, midbrain, hindbrain, and the spinal cord along the anterior-posterior axis [15] (reviewed in Green et al. 2014 [16]). At the neural groove stage, the area between the presumptive forebrain and the midbrain is patterned by an interplay between Wnts and anterior neural boundary-derived Wnt inhibitors patterns. During the early neural tube stage, the forebrain is divided into the telencephalon at the anterior part and the diencephalon at the posterior part. In addition, the boundary between the midbrain and the hindbrain is formed by an apparent constriction of the neural tube (the isthmus). Dorsoventral patterning is formed by gradients that are perpendicular to the rostro-caudal axis, and Shh signaling plays an important role in this process (reviewed by Ribes and Briscoe, 2009 [17]). Wnts are fundamental for neurogenesis and cell fate determination from early brain development through the adult stage (reviewed by Noelanders and Vleminckx, 2016 [18]). For example, Wnts are essential for the posterior forebrain induction and patterning of chick embryos [19] and the midbrain and isthmus development of *Xenopus* embryos [20]. The temporal and the spatial expression patterns of members of Wnt and BMP genes are similar in humans and in mice, suggesting that secondary organization centers, such as the cortical hem, have similar roles [21]. It has been recently demonstrated that human hESCs grown in microfluidic devices and subjected to WNT gradients exhibited rostro-caudal identities from forebrain to midbrain to hindbrain, including the formation of isthmic organizer-like characteristics [22]. More attempts have been done to generate dorsoventral patterning. These include fabricated organoid-on-chip microfluidic devices [23,24], addition of signaling-molecule-secreting cells to the culture [25], and fusion of different types of organoids [26,27,28]. Spontaneous polarization of spinal cord organoids was observed in rotary cultures, where the organoids elongated as their differentiation progressed [29].

Classical developmental studies of chick embryos involve the insertion of morphogen-soaked beads directly to the developing tissue [30,31]. The morphogens diffuse from the beads (the signaling source) to the tissue and initiate gene expression changes. Within the tissue, the concentration of the morphogen is decreased as the distance from the bead is increased resulting in gradient formation [30]. Morphogen-loaded beads (WNT3a) were found to affect ESCs divisions in vitro [32], but are yet to be used in organoid research.

In this report, we used beads soaked with either BMP4 or a WNT agonist to influence the spatial identities within brain organoids that were derived from hESCs. To follow their development, we developed a system for live-imaging of the organoids and beads co-cultures.

We modified our previously published device, in which early brain organoids were confined on the top by a membrane and on the bottom by a glass coverslip [6]. The device allows in situ imaging, but also restricts the growth in the *z*-axis. The limited thickness improves the nutrient exchange and is beneficial for the imaging resolution. The inserted hESC aggregates self-organize into neuroepithelium (NE) structures, and express genes that are typical to radial glia cells [6]. However, since this system is sealed, it has not been possible to access the tissue for performing manipulations during the experimental timeline, such as inserting and removing beads. 

First, we describe the technical details of this novel fabricated device and its application in growing and live-imaging of human brain organoids. We demonstrate the ability to co-culture organoids with morphogen-soaked beads and to freely access the tissue. We then show that embedding morphogen-soaked beads in close proximity to an organoid result in distinct morphologies and gene expression changes. We used this system to study the role of WNT and BMP4 gradients in establishing spatial identities in human brain organoids. 

By embedding CHIR99021 (CHIR, WNT agonist)- and BMP4-soaked agarose beads adjacent to the growing brain organoid, we were able to induce dorsal and ventral transcription factors in the area that was close to the beads versus the far side, respectively. By increasing the concentrations of CHIR and BMP4 on the agarose beads, we were able to modulate the expression of rostral and caudal transcription factors in the early brain organoids. We observed a dose-responsive change in the expression of telencephalon-related genes, including medial pallium and cortical-hem markers; dorsal and ventral midbrain markers; and isthmus-related markers. These findings suggest that our approach may provide the possibility to generate spatial cues within organoids, and an experimental platform to study the effects of modulating WNT/BMP-mediated signaling.

## 2. Materials and Methods 

### 2.1. Cell Lines

We used early passages of the NIH-approved embryonic stem cell line NIHhESC-10-0079 (WIBR3), which were stably labeled with a fluorescent H2B-mCherry reporter, labeling histones, and a LifeAct-GFP reporter labeling actin, as previously described [6] and maintained in NHSM media [33,34]. Cell transfection was carried out in a NEPA21 electroporation system according to the company’s protocol (Nepa Gene, Chiba, Japan). The FUCCI-reporter lines were generated using PiggyBac-based plasmids, that were stably integrated into the cell genome at random sites [35]. FUCCI reporter plasmids were received from Dr. Atsushi Miyoshi: mCherry-hCdt1 GeneBank, AB512478, mVenus-hGeminin, GeneBank, AB512479 [36], and were subcloned into pCAGGS—PiggyBac plasmids as previously described [37]. We co-electroporated pCAG:hCdt-H2B-mCherry and pCAG:hGemininH2B-GFP plasmids together with a pCAGGS-PBase plasmid expressing the PiggyBac transposase [35]. The constructs do not interfere with the cell cycle, as they consist of a fragment of these proteins (amino acids 1–110 of hGeminin and 30–120 of hCdt1) [36]. Following electroporation, cells were plated, passaged once after several days, and selected by FACS to ensure for high levels of expression.

### 2.2. Human ESC Aggregates

Human ESCs were cultured on Matrigel™ (#354234, Corning, New York, NY, USA) coated plates using naïve human stem-cell media (NHSM) as previously described [33,34]. The NHSM media contained DMEM/F12 (21331-020 Gibco, ThermoFisher, Waltham, MA, USA), Albumax I (5gr, 11020-039, Invitrogen, Carlsbad, CA, USA), Pen-strep (5 mL, 03-031-1B, Biological Industries, Beit HaEmek, Israel), L-glutamine (5 mL, 02-022-1B, Biological Industries, Beit HaEmek, Israel), NEAA (5 mL, 01-340-1B, Biological Industries, Beit HaEmek, Israel), KSR (Knockout Serum Replacement, 50 mL, 10828-028, Invitrogen, Carlsbad, CA, USA), human insulin (12.5 μg/mL I-1882, Sigma, St. Louis, MO, USA), Apo-transferrin 100 μg/mL T-1147 Sigma, St. Louis, MO, USA), Progesterone 0.02 μg/mL P8783 Sigma, St. Louis, MO, USA), Putrescine 16 μg/mL P5780, Sigma, St. Louis, MO, USA), Sodium selenite 30 nM S5261 Sigma, St. Louis, MO, USA), 2-mercaptoethanol 50 mM (50 μL, 31350-10 Gibco, ThermoFisher, Waltham, MA, USA), L-ascorbic acid 2-phosphate (Vitamin C, 50 μg/mL, A92902, Sigma, St. Louis, MO, USA), Human LIF (20 ng/mL, Homemade), FGF2 Recombinant human (FGF-basic; 8 ng/mL, 100-18B, Rocky Hill, NJ, USA), TGFB1 (1 ng/mL, 100-21c, Peprotech, Rocky Hill, NJ, USA), IWR1 (5 μM, HY-12238, Medchem Express, Monmouth Junction, NJ, USA), Chir99021 (3 μM, HY-10182, Medchem Express, St. Louis, MO, USA), ERKi (PD0325901, 1 μM, HY-10254, Medchem Express, Monmouth Junction, NJ, USA), p38i (BIRB0796, 2 μM, HY-10320, Medchem Express, USA), JNKi (SP600125, 5 μM, 1496, Tocris, Bristol, UK), BMPi (LDN193189, 0.4 μM, HY-12071A, Medchem Express, Monmouth Junction, NJ, USA) and ROCKi (Y27632, 5 μM, HY-10583, Medchem Express, Monmouth Junction, NJ, USA). The Matrigel-coated plates were prepared as previously described [34]. Briefly, 10 cm tissue-culture plates were coated with 1:100 Matrigel diluted in DMEM/F12 (21331-020 Gibco, ThermoFisher, Waltham, MA, USA) and kept at 37 °C for at least 2 h before use.

When the hESC colonies were visible to the eye, the cells were dissociated using Trypsin-EDTA (0.05%, ThermoFisher, Waltham, MA, USA), and diluted in serum-containing media to inhibit trypsin activity. Cells were centrifuged and re-suspended in NHSM with 40 μM Rho Kinase inhibitor Y-27632 (MCE, Monmouth Junction, NJ, USA). The cells were counted and diluted to a final concentration of 1.5 × 10^4^ cells/mL. About 3600 cells were dispensed into ultra-low cell-attachment 96-plate (S-BIO, Hudson, NH, USA). Within several hours, the cells aggregated at the bottom of the wells. To generate brain organoids with NE domains, Neural induction media [6] was added 24 h from aggregation. The neural induction media contained DMEM/F12 (100 mL, Gibco, 21331-020, ThermoFisher, Waltham, MA, USA), N2 supplement 100× (1 mL, Gibco, 17502-048, ThermoFisher, MA, USA), Glutamax (1 mL, Gibco 35050-038, ThermoFisher, Waltham, MA, USA), MEM-NEAA 100× (1 mL, 01-340-1B Biological Industries, Beit HaEmek, Israel), Pen-Strep (1 mL, 03-031-1B Biological Industries, Beit HaEmek, Israel), and Heparin solution (1 ug/mL, H3149 Sigma, St. Louis, MO, USA).

### 2.3. Device Assembly and Hydrogel Embedment

Day 7 cell aggregates were collected by a pipette and suspended in fresh media. Next, drops of 4 μL collagen-laminin-based hydrogel (100% Matrigel) were placed on an imaging coverslip which was on the bottom of the fabricated device. Each organoid was embedded individually in a drop. Nine to twelve aggregates were embedded per device. Then, the device was inserted into the incubator for 25 min for gelification. Next, 1 mL of media was added on top of the organoids. The grid insert, which supports a semi-permeable polycarbonate membrane (Whatman^®^ Nuclepore Track-Etched Membranes, pore diameter 0.1 μm), was placed on top of the aggregates, and two nylon bolts were screwed to seal the device. Illustrations of the device assembly are found in Appendix A. The device was then filled with neural differentiation media supplemented with 1:1 DMEM/F12 (100 mL 21331-020 Gibco, ThermoFisher, Waltham, MA, USA) and Neurobasal (100 mL, 21103-049 Gibco, ThermoFisher, Waltham, MA, USA), N2 supplement 100× (1 mL, Gibco, 17502-048, ThermoFisher, Waltham, MA, USA), B27-Vitamin A 50× (2 mL, 12587-010 Gibco, ThermoFisher, Waltham, MA, USA), Insulin solution human 1000× (100 μL, I9278 Sigma, St. Louis, MO, USA), 2-beta mercaptoethanol (50 mM, 10 μL, 31350-10 Gibco, ThermoFisher, Waltham, MA, USA), Glutama× (2 mL, 35050-038 Gibco, ThermoFisher, Waltham, MA, USA), MEM-NEAA 100× (1 mL, 01-340-1B Biological Industries, Beit HaEmek, Israel), Pen-Strep (2 mL, 03-031-1B Biological Industries, Beit HaEmek, Israel), 20 ng/mL EGF (PeproTech, IL) and 20 ng/mL FGF2 (PeproTech, Rocky Hill, NJ, USA) [6] and kept in a cell incubator at 37 °C and 5% CO_2_. Media was changed every other day. 

### 2.4. Beads Soaking with Morphogens

Affi-Gel blue-gel beads (BioRad, Hercules, CA, USA) were loaded with the proteins or small molecules as previously described [38,39]. Briefly, beads were washed with PBS several times and incubated with the morphogens for 1 h in 37 °C. For brain organoids in 1× condition, we used 3 μM CHIR99021 (GSK3 inhibitor; Stemgent, Cambridge, MA, USA) and 0.5 nM BMP4 (R&D, Minnesota, MN, USA). For 2× condition, we used 6 μM CHIR99021 and 1 nM BMP4. For 4× condition, we used 12 μM CHIR99021 and 2 nM BMP4. For controls, the beads were soaked in the media. Protein-soaked beads were stored at 4 °C and used within one week. Prior to the insertion into the devices, the beads were washed with PBS. Each bead was placed in close proximity to an organoid in the same hydrogel drop. After 3 days, the beads were carefully removed and replaced with fresh beads. 

### 2.5. qPCR

Prior to RNA collection, organoids were dissected in half. The two sides of the organoids were collected separately. Total RNA was extracted using the RNeasy Mini kit (Qiagen, Hilden, Germany) following the manufacturer’s protocol and followed by DNAseI treatment. RNA concentration was measured using Nanodrop (Thermo Scientific, Waltham, MA, USA). cDNA was synthesized using M-MLV reverse transcriptase (M3682, Promega, Wisconsin, WI, USA). Real-time reactions were performed in triplicates using KAPA SYBR FAST qPCR Kit (2×) on Quant Studio 5 Real-time PCR system (Bio-Rad, Hercules, CA, USA) according to manufacturer’s recommendation. Expression levels were normalized against GAPDH using the ΔΔ threshold cycle (Ct). Primer sequences [7,40] are found in Appendix A. 

### 2.6. Immunostainings 

Organoids were fixed using 4% PFA for 4 h at RT and washed thoroughly with PBS. Whole organoids or cryosections of 20 μm thickness were permeabilized for 5 min three times using 0.1% Triton X-100 and then treated with DNase1 for 15 min and blocked in blocking solution (PBS, 0.1% Triton X-100, 10% HS; 10% FBS) for 60 min. Primary antibodies were incubated in blocking solution overnight at 4 °C. After three washes with 0.1% Triton X-100 appropriate secondary antibodies (1:200, Jackson ImmunoResearch, West Grove, PA, USA) were diluted in blocking solution and incubated for 2 h at RT. Slices were mounted onto glass slides using ProLong Gold Antifade Mountant (Thermo Fisher Scientific, Waltham, MA, USA) and imaged.

### 2.7. Antibodies

Rabbit anti-PAX6 (previously Covance #PRB-278P, 1:50, BioLegend, San Diego, CA, USA); Mouse anti-beta-catenin (BD biosciences, San Jose, CA, USA, #610153, 1:200); Rabbit anti-AQP1 (ab15080, 1:200, Abcam, Cambridge, UK). Secondary antibodies (1:200, Jackson ImmunoResearch, West Grove, PA, USA): Donkey Anti-Mouse IgG Alexa Fluor 647 #715-605-150; Donkey Anti-Rabbit IgG Alexa Fluor 647 #711-607-003. 

### 2.8. Imaging and Analysis 

Imaging was carried out in a Dragonfly high speed confocal microscope (Andor, Belfast, Northern Ireland). Imaging was performed in Fusion 2.0 software. Image analysis was performed in FIJI. The statistical analysis was performed in GraphPad Prism 7 by Student’s *t*-test or ANOVA. For One-way ANOVA, post-hoc Tukey’s multiple comparisons test was used. All variances were similar. Error bars represent S.E.M.

## 3. Results

### 3.1. Efficient Live-Imaging of Neuroepithelium Tissues Derived from hESCs

To follow the development of human neuroepithelium (NE) organoids [1,3,6], hESC aggregates were embedded in drops of collagen-laminin-based hydrogel (Matrigel™) and inserted into a micro-fabricated compartment (h = 250 ± 10 µm, Figure 1A and Appendix A; details of the device assembly are found in the methods section). For visualization, we used hESCs which were genetically labeled with LifeAct-GFP (labeling the actin cytoskeleton) and H2B-mCherry (histone labeling the nuclei). During the first few days, several NE structures emerged (Figure 1B and Appendix A), where the cells stretched from the inner (apical) surface to the outer (basal) surface (Figure 1B’). The proliferating NE centers contained PAX6^+^ radial glia cells (Appendix A), indicating successful neural induction. Within two weeks on-chip, the organoids reached a size of about a millimeter in diameter (Figure 1C, the area of the organoids measured 0.8  ±  0.04 mm^2^ and the thickness was up to 250 μm), while the tissue was still confined by the coverslip and the membrane. Opening and restoring the insert unit of the compartment did not interfere with the growth of the tissue.

To follow the cell-cycle within the NE domains, we aggregated hESCs which were labeled with FUCCI markers: hGeminin-GFP (a S-G2- and M-phase marker) and hCdt1-mCherry (a G1-phase marker) [36]. Red nuclei represent cells that are in G1-phase, green nuclei are in S-phase or G2/M-phases (darker nuclei), and yellow nuclei that express both reporters are in G1-to-S phase transition (Appendix A). We then followed the proliferation of the different domains at day 17 in an overnight time-lapse movie (18 h). The averaged percentage of cells in the different cell cycle phases was: 32.57% ± 7.17% cells in G1-phase (Red); 58.47% ± 8.09% cells in S-phase (bright green); 4.64% ± 1.65% cells in G2/M-phases (large nuclei, dark green); and 4.31% ± 2.57% cells in G1-to-S-phase (yellow). For comparison, in cortical progenitors of E12.5 mice embryos, 32.4% cells are in G1-phase and 48% are in S-phase [41]. The data were normally distributed, indicating that the proliferation rate was similar in the distinct proliferative domains.

To check whether the percentages of the specific cell cycle stages change during live-imaging, we followed the percentage of the cells in the different cell cycle phases in for 18 h in 3 h intervals. No changes were found in the %S-, %G1-to-S-, %G2/M-, and %G1-phases (Appendix A), indicating that during the live-imaging, the percentages of the specific cell cycle stages, remained constant.

We observed that the cell nuclei were radially oriented, and performed an up-and-down radial motion as the cell cycle progressed, with the cell division taking place in the inner (apical) surface (see large green nuclei within the inner dashed line in Appendix A). These recapitulate a key feature of the developing ventricular zone in vivo, where neuronal progenitors preform interkinetic nuclear migration, coupled with the cell cycle [42]. Taken together, we have successfully grown and imaged human early brain organoids, with the opportunity to access the tissue at any time. From early stages, we observed the formation of proliferating PAX6^+^ NE domains, which featured interkinetic nuclear migration as the cell-cycle progressed.

### 3.2. Different Concentrations of CHIR and BMP4 Polarize Human Brain Organoids

In vivo, the topography of a tissue and the cellular identities within it are determined by temporal and regional signaling activities (reviewed by Sagner and Briscoe, 2017 [11]). In floating organoid cultures, it is likely that these gradients are not formed in a consistent fashion. Furthermore, organoids lack a reproducible topographic organization, and the spatial identities are generated stochastically within the cell mass. In the past, ectopic formation of gradients of morphogens were introduced in chick embryos by the addition of morphogen-soaked beads [30,31]. We therefore adapted this method to generate distinct cell identities in organoids.

Gradients of Wnt/BMPs [43] and Shh [44] induce dorsal and ventral fates in chick embryos, respectively (reviewed by Lee and Jessell, 1999 [45], and later by Wilson and Maden, 2005 [46]), and increasing concentrations of Wnt are found in the anterior-posterior axis (reviewed in Green et al. 2014 [16]). Here, we explored the WNT/BMP-dependent patterning of human brain organoids [6]. We investigated whether the application of CHIR99021 (CHIR; GSK3i, WNT agonist) and BMP4, by beads in close proximity to the organoids could pattern the developing tissue. We hypothesized that the cells that are in close proximity to the bead area sense a higher concentration of the introduced substances, similar to the gradients that are induced by the beads in vivo. The local differences in concentrations of the added substances may result in the establishment of a dorso-ventral patterning within each organoid, therefore mimicking local signaling centers. In parallel, we investigated whether applying increasing concentrations of CHIR and BMP4 on the beads could change the gene expression pattern in the AP or rostro-caudal axis, from the forebrain to the hindbrain. In these experiments we used the on-chip device described above that allows the organoids to retain a specific position, easy access and the possibility to perform live imaging.

As a baseline condition (referred to as 1× concentration), we used 3 μM CHIR and 0.5 nM BMP4, which were sufficient to generate cortical hem and choroid plexus domains in floating cultures of forebrain organoids [7,47,48]. For the increased conditions, we used 2× concentration (6 μM CHIR and 1 nM BMP4) and 4× concentration (12 μM CHIR and 2 μM BMP4). The increasing concentrations were chosen arbitrarily, and were aimed to test whether changing the concentrations of the CHIR and BMP4 could influence the AP patterning (forebrain to hindbrain) which is found in vivo. The beads were embedded in close proximity to the aggregates in the same hydrogel drop on day 7. No additional CHIR and BMP4 were supplied to the media. We define the “bead area” as the half of the organoid, which is closer to the beads, and the “far side” as the opposite half. These two areas were collected separately and analyzed gene expression by qPCR. The experimental timeline is found in Figure 2A. Examples of organoids with the beads on day 7 and on day 14 are found in Figure 2A’.

To investigate whether the morphogens generate gradients of gene expression within each organoid, we examined the expression of the beta-catenin protein, that is a downstream effector of the WNT pathway (Figure 2B and Appendix A). We observed high levels of beta-catenin expression in close vicinity to the bead and low levels of beta-catenin immunoreactivity in the far side, indicating that the molecules introduced to the beads have the capability of generating expression gradients within the organoids.

We then measured the gene expression changes following the addition of CHIR- and BMP4-soaked beads using this baseline concentration. In the developing mouse brain, the main source of WNT and BMP is the cortical hem, which instructs the development of the hippocampus and the choroid plexus, all part of the dorsal telencephalon [49,50,51,52]. FOXG1, a known antagonist of BMPs [53], is expressed in the telencephalon from the medial pallium to the neocortex, but not in the choroid plexus (ChP) and the hem [54]. LEF1, which is directly regulated by WNT and regulates the generation of dentate gyrus, is expressed in a gradient from the medial pallium, but is not found in the dorsolateral pallium [55]. PAX6 is expressed in the proliferative domain of the cortex [56]. We therefore defined the medial pallium by the expression of LEF1, PAX6, and FOXG1 (LEF1^+^/PAX6^+^/FOXG1^+^), while the dorsal part lacks LEF1 expression (LEF1^-^/PAX6^+^/FOXG1^+^) as has been previously designated [7] (summarized in Figure 2C. Reviewed also by Mallamaci and Stoykova, 2006 [57]).

Under the baseline condition, the levels of *FOXG1* and *PAX6* were increased in both halves of each organoid, indicating that the CHIR + BMP4 treatment was able to increase the expression of these telencephalic-related genes in comparison to the untreated controls (Figure 2D). The levels of *LEF1* were decreased in the bead area, indicating a more dorsal fate (Figure 2D). The levels of *OTX2*, a general marker for neural progenitors, remained unchanged (Appendix A). The expression of the cortical hem and ChP marker *LMX1a*, significantly increased in the bead area (Figure 2E). The levels of *WNT3a*, that is secreted by the hem and is crucial for hippocampal development [49,50], were elevated in the bead area (Figure 2E). *AQP1*, which is uniquely expressed in the ChP, was also upregulated in the bead area (Figure 2E). Taken together, at the baseline concentration both sides of the organoids were enriched in the telencephalon marker *FOXG1* in comparison to the control brain organoids, nevertheless it is noted that an internal gradient was established, and cortical-hem and ChP-related genes which were observed to be enriched in floating cultures [7,8] were upregulated in the bead area.

### 3.3. Liquid-Filled Cysts and Larger Ventricles Correspond to An AQP1 Elevation in the 1× Condition

Complementary to the elevation in the *AQP1*, we observed the spontaneous formation of clear cysts in the bead area (Figure 3A). In another study, organoids that were treated with similar concentrations of CHIR and BMP4 developed cysts that contained cerebrospinal fluid (CSF), known to be produced by the ChP [8].

Interestingly, live-imaging of the developing organoids revealed that the ventricles of the CHIR + BMP4 condition organoids were enlarged in comparison to the controls (Figure 3B, enlarged panels i-ii; Extended panel of the developing organoids is found in Appendix A). Correspondingly, we detected high levels of AQP1 in the apical (inner) surface of the 1× condition NE domains, in comparison to the controls (Figure 3C). Taken together, in the 1× condition, the bead area features liquid-filled cysts, and the water content within the ventricles of the organoids was increased.

### 3.4. Dorsal and Ventral Midbrain-Related Markers Are Upregulated in the 2× Condition

Next, we measured the gene expression changes in the 2× concentration condition. Here, we noted the emergence of midbrain-related genes. In mouse embryos, the midbrain is patterned by a combination of WNT signaling from the roof plate (RP) and the floor plate (FP) [58,59] and SHH signaling from the FP [60]. *Pax6* is expressed in the dorsal side of the midbrain, while *Foxa2* and its downstream targets, *Nkx6.1* and *Ngn2*, are expressed in the ventral domain (Figure 4A) (reviewed by Gale and Li, 2008 [61]). *Lmx1a*, which is directly upregulated by WNT, is expressed in the RP, and later emerges in the ventral side (reviewed by Joksimovic and Awatramani, 2014 [62]).

Similar to the baseline condition, we observed an elevation in the levels of *AQP1* in the bead area (Figure 4B). Consistent with the increase in the CHIR and BMP4, there was an overall reduction in the telencephalon marker *FOXG1*, in particular in the bead area (Appendix A, late cycle). The levels of *LEF1* and *OTX2* were unchanged (Appendix A). In the bead area, the dorsal-side markers *LMX1a* and *PAX6* were elevated (Figure 4C). In contrast, the expression of the FP-related genes, *FOXA2*, *NKX6.1,* and *NGN2*, was enriched in the far side of the organoid (Figure 4D,E). We therefore conclude that the 2× condition induced the expression of midbrain-related genes. Specifically, the bead area was enriched in dorsal/RP-related markers, while in the far side in ventral/FP-related markers were expressed.

### 3.5. Midbrain-Hindbrain Border Domains and Isthmus-Related Markers Are Observed in the 4× Condition

Finally, we analyzed the 4× concentration condition. The isthmus, that is located at the midbrain-hindbrain border, is a key organizer for the midbrain as well as the cerebellum (reviewed by Joyner, Liu and Millet, 2000 and in Nakamura et al., 2005 [63,64]). During mouse brain development, the isthmus is the source of FGF8 signaling, and serves as a boundary between the *Otx2*-expressing midbrain and the *Gbx2*-expressing hindbrain [65,66,67]. Other midbrain-hindbrain markers include the secreted molecule *Wnt1*, and the transcription factors *Pax2* (at the posterior midbrain), *En1,* and *En2* (Figure 5A) (reviewed by Joyner, Liu and Millet, 2000, and in Nakamura et al., 2005 [63,64]). In another research, midbrain-specific differentiation from hESCs by activation of Wnt signaling was characterized by the elevation of *EN1*, *EN2*, *GBX2*, *PAX2,* and *FGF8* expression levels [68]. Here, we observed a general increase in the levels of the posterior midbrain marker *PAX2* in the treated organoids, with a further elevation in the far side of the organoids (Figure 5B). Similarly, the expression of the midbrain-hindbrain border genes, *FGF8*, *EN1* and *EN2*, was also elevated, with a further increase in the far side (Figure 5C). The FP-related markers, *WNT1*, *NKX6.1,* and *NGN2*, were increased in the far side of the organoid (Figure 5D). The levels of *FOXG1* and *GBX2*, that indicated forebrain and cerebellar identities, correspondingly, were below detection levels. Complementary to these results, no changes were observed in the levels of *PAX6*, *OTX2*, *LMX1a,* and *AQP1* (Appendix A). Collectively, these results indicate that the 4× concentration condition results in the expression of isthmus-border posterior markers, where the far side of the organoid is enriched in ventral-related genes.

Taken together, we were able to influence gene expression within the organoids at two levels: First, the whole organoid was posteriorized/caudalized as we increased the CHIR and the BMP4 concentrations on the beads. Second, in the different experimental conditions, an internal gradient within each organoid was established, where the bead area was characterized by the expression of dorsal/RP-related genes, and the far side was characterized in the upregulation of ventral/FP-related genes.

## 4. Discussion

Organoids are three-dimensional (3D) cell cultures that mimic key aspects of in vivo organ development, from stem cells through complex tissues. Growing human organoids is an emerging research tool to study basic developmental processes in a human background. Traditionally, organoids are grown in floating cultures. This method poses two main challenges: First, the thick architecture of the organoids limits the live-imaging capacity. This issue was previously addressed [6] and further examined in this report. Second, embryonic tissues in vivo are exposed to gradients of developmental signals and spatial cues (morphogens) [30], which are usually not present in vitro. One reason is that the fate-patterning molecules are added uniformly to the media. This also contributes to a stochastic organization of the developing organoids. Ideally, organoid models should also reflect the asymmetries of the developing brain, such as the topography of the dorsoventral, mediolateral, and anteroposterior axes.

One strategy to generate areal organization within organoids has been by embedding morphogen-secreting cells in a pole of an aggregate [25]. Here, we adapted a traditional method from classical developmental in vivo studies, in which morphogen-soaked beads were inserted directly into the developing tissue [30,31]. The beads are interpreted as local organizers and induce distinct cell fate changes. The organoids and the beads co-cultures were grown on-chip, and followed by live imaging.

In the first part of this study, we tested our devices for their ability to support the growth of the organoids and the possibility to use them for live imaging. In just a few days, the organoids displayed multiple organized proliferative centers, in which the PAX6^+^ radial glia progenitors exhibited radial orientation and performed interkinetic nuclear motion which was coupled to the cell cycle. These features recapitulate previous observations in animal models [42] and in organoids [6]. Within a few weeks, the organoids had reached a millimeter in diameter. During this time-course, the developing organoids were followed by live imaging. We conclude that our on-chip method is a viable platform to grow and to image 3D organoids.

We next explored the role of WNT and BMP4 gradients in establishing dorsoventral (DV) and anteroposterior (AP) fates in human brain organoids. Increasing gradients of WNTs and BMPs exist in the developing brain from the anterior/rostral side to the posterior/caudal side. Additionally, a local concentration gradient in the DV axis exists, where WNTs and BMPs are expressed more at the dorsal side.

Briefly, studies in *Xenopus* [69], zebrafish [66], and chick embryos [15] have demonstrated an essential role for WNT signaling in AP patterning. It was further suggested that orthogonal gradients of WNTs and BMPs can pattern intersecting axes in vertebrates, similar to their function in the *Drosophila* imaginal disc [69]. A current consensus is that posteriorizing signals establish broad AP domains, in which secondary organizing domains are formed (reviewed by Green et al. [16]). These further refine the patterning within each gross domain. It is therefore suggested that WNT and BMPs can function both as broad AP patterning molecules, and as local DV-patterning signals. To study the mechanism of transport of morphogens, recent synthetic approaches were used [70,71]. Interestingly, the synthetic morphogens formed gradients, which demonstrated a high concentration near the source but also a low concentration far from the source. In the telencephalon, the main source of WNT and BMP is the cortical hem, a regulator of the hippocampus development [49]. The dorsal midline, which gives rise to the choroid plexus, is an additional source for BMP [51]. A more posterior source for WNT and BMPs is found at the midbrain-hindbrain boundary, the isthmus (reviewed by Nakamura et al., 2005 [63]). The isthmic organizer regulates the development of the midbrain and the anterior hindbrain (reviewed by Wurst and Bally-Cuif, 2001 and in Nakamura et al., 2005 [63,72]). The strategy of growing co-cultures of organoids and morphogen-soaked beads, which is presented here, provides us the opportunity to study how different concentrations of WNT and BMP4 affect the DV and the AP patterning in a human background. A different approach to generate an organizing center has been done using a cluster of cells expressing SHH [25]. Our approach has the possible advantage that it is a short-term treatment and most likely the substances are active for a period of approximately three days. Remarkably, the treatment with the two factors, CHIR (WNT agonist) and BMP4, on agarose beads, had two major consequences: First, within each organoid, dorsal-related markers emerged in close vicinity to the bead, while the expression of ventral-related genes was higher in the far side of the organoid. Second, increased concentrations of CHIR and BMP4 were able to generate more posterior/caudal identities (summarized in Figure 6A). Our method induced three types of brain organoids. At the baseline concentrations, the expression of dorsal forebrain-related genes emerged. We detected dorsal midline and cortical hem characteristic markers in the bead area, whereas ventral medial pallium domains (*LEF1^+^/PAX6^+^*) were enriched in the opposite side of the organoid. These findings recapitulate the cortical hem- and ChP-like identities, which were generated by applying similar concentrations of CHIR and BMP4 uniformly to the media [7]. Clear cysts formed close to the bead area, where the expression of *AQP1,* a gene encoding for a water-specific channel, was elevated. In another study, similar concentrations of CHIR and BMP4 resulted in CSF-containing protrusions in floating organoids [8]. Interestingly, the ventricles of the developing NE were also enlarged, and the adjacent cells expressed AQP1.

An additional increase in the concentration of CHIR and BMP4 resulted in the expression of midbrain-related markers, thus suggesting a more posterior/caudal identity. Within the organoids in the bead area we noted the expression of dorsal midbrain-related markers (*LMX1a^+^/PAX6^+^*), whereas in the far side we noted the emergence of FP-related markers (*FOXa2^+^* and its downstream targets, *NKX6.1^+^* and *NGN2^+^*).

Finally, the highest concentration tested resulted in the expression of markers that are involved in patterning of the mesencephalon. We detected the expression of *FGF8*, *WNT1*, *EN1/2,* and *PAX2*, all of which are required for midbrain and cerebellar development (reviewed by Joyner, Liu and Millet, 2000 and in Nakamura et al., 2005 [63,64]). Overall, the differential gene expression observed in the two halves of the organoid using the three conditions suggests that the cells of the organoids perceived the morphogens in relation to their distance from the beads, possibly as a gradient. Moreover, the organoids were posteriorized by increasing the concentrations of CHIR and BMP4 on the beads. It has been recently demonstrated that a WNT gradient is sufficient to induce AP cell fates [22], therefore it may be possible to postulate that the changes in the dorsal-ventral axis requires BMP or the combination of BMP and WNT. The changes in gene expression that were observed following the three treatments are summarized in Figure 6B.

In neurodevelopmental diseases such as autism spectrum disorder and schizophrenia part of the pathophysiology is attributed to altered regional specification and defective connectivity. Therefore, achieving reproducible topographic organization in human organoids and generating multi-regional domains in a single system are currently an emerging field. Our protocol provides an opportunity to pattern the developing tissues in a more refined manner and to perform live-imaging. In future studies, it will be important to examine the extent to which the polarity can be maintained during long-term growth.

The emergence of DV and AP topographies in human brain organoids is a key point in our research. By using this method, we emphasize the role of WNT/BMP signaling in patterning the DV and AP axes. Finally, we hope that this application will be used in future organoid studies to generate regional identities resembling the developing human brain.

## 5. Conclusions

We have developed a novel on-chip platform to follow the development of human brain organoids with live imaging for weeks. We have adapted the use of morphogen-soaked beads, a classical in vivo technique to study basic developmental processes, to human organoid research. Applying increasing concentrations of CHIR (WNT agonist) and BMP4 on the beads has resulted in the emergence of dorsoventral and rostrocaudal/anteroposterior identities in human brain organoids. These results elaborate on the WNT/BMP-dependent patterning of the DV and the AP axes in human brain organoids.

## Figures and Tables

**Figure 1 bioengineering-07-00164-f001:**
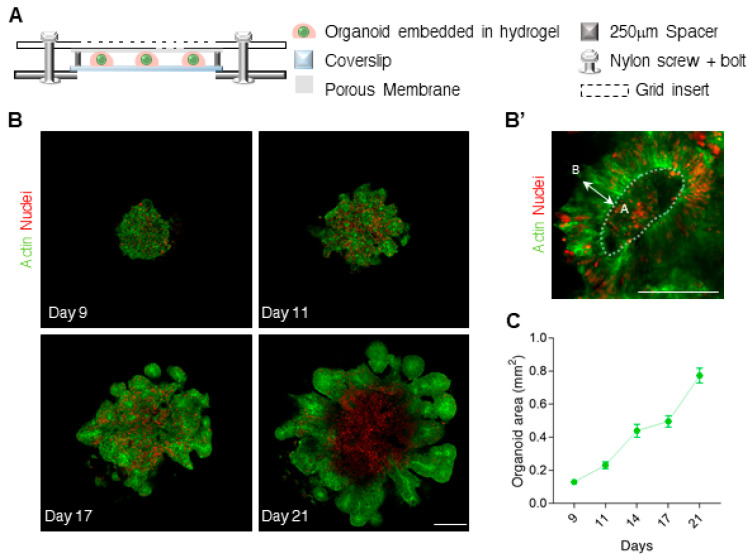
Live imaging of Neuroepithelium organoids grown on chip. (**A**) An illustration of the assembled organoid compartment (side view), consisted of a removable grid insert with a membrane (top), a spacer (h = 250 mm) and a glass coverslip (bottom). (**B**) Fluorescent images showing the development of a human brain organoid, and the emergence of neural epithelium (NE) centers. The cells are labeled with a membranal marker (LifeAct-GFP, Green) and with a nuclear marker (H2B-mCherry, Red). (**B’**) Enlarged image of a NE center. Dashed line marks the inner/apical (A) surface. The cells within the NE are starched between the outer/basal (**B**) and the apical side. Cell divisions occur on the apical side. (**C**) Organoid area (mm^2^) over time. N = 32 organoids. Scale bars 200 μm. Error bars represent ± SEM.

**Figure 2 bioengineering-07-00164-f002:**
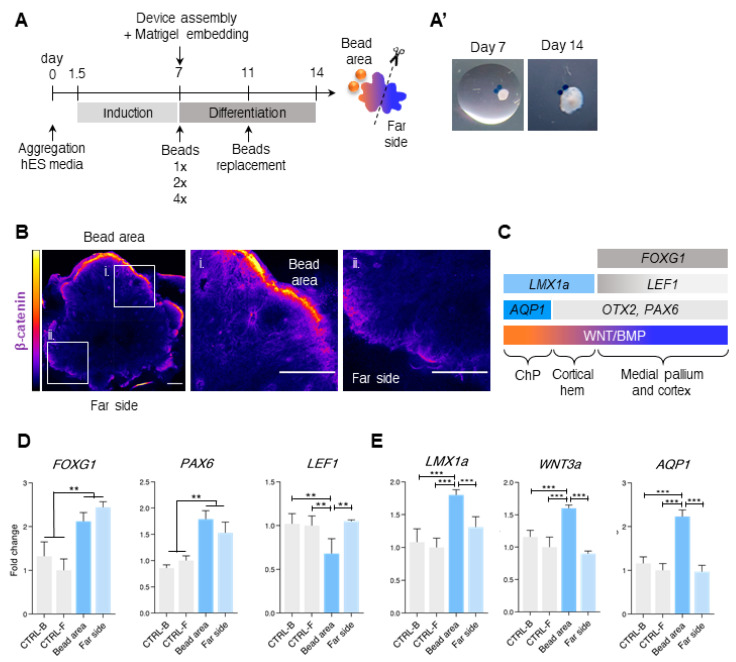
Baseline concentrations of CHIR and BMP4 on beads induce telencephalon-related identities. (**A**) Experimental timeline for co-culturing brain organoids and beads. The beads were soaked in the media (for control) or in different concentrations of CHIR99021 and BMP4. Fresh beads were applied at day 7 and replaced with fresh beads at day 11. At day 14, the organoids were dissected and collected as shown. (**A’**) Images showing an organoid at Day 7 (insertion to the device) and at day 14 (collection). (**B**) Immunohistochemistry and heatmap of beta-catenin expression (yellow-high; blue-low) at day 11 of the 1× condition. (i,ii) Enlarged captions of (i) the bead area and (ii) the far side. (**C**) Schematic gene expression in the medial pallium, cortical hem, and choroid plexus (ChP) during development. (**D**) Gene expression changes of the telencephalon gene *FOXG1*, and the medial pallium genes *PAX6* and *LEF1* which were measured by qPCR at day 14 following 1× treatment. (**E**) Gene expression of *LMX1a*, *WNT3a,* and *AQP1* which were measured by qPCR at day 14 following 1× treatment. Error bars represent ± SEM. N = 24 organoids per experimental group. Comparisons were analyzed using *ANOVA* with post-hoc Tukey’s multiple comparisons test (DF = 20): n.s. non-significant *p*-value > 0.05, * *p*-value < 0.05, ** *p*-value < 0.01, *** *p*-value < 0.001. Scale bars 200 μm.

**Figure 3 bioengineering-07-00164-f003:**
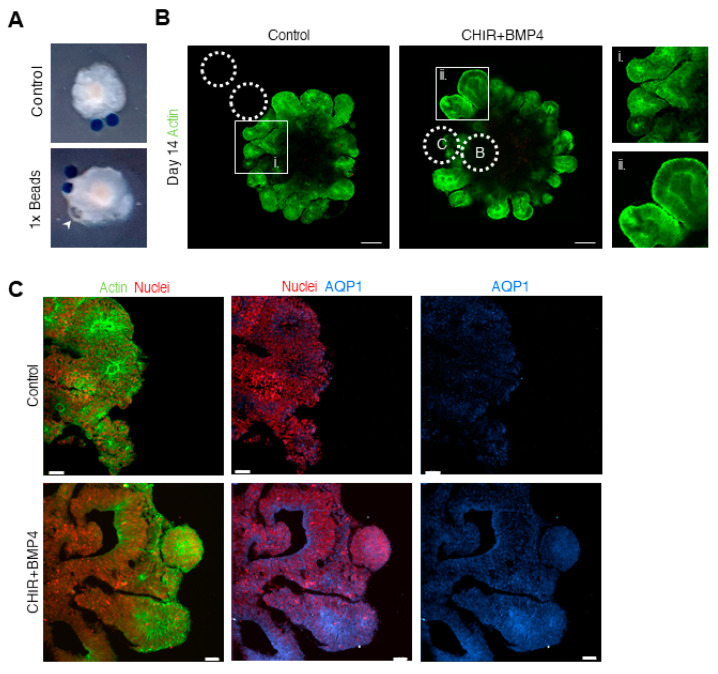
Liquid-filled cysts and larger ventricles are observed in the 1× condition. (**A**) Day 14 control and 1× condition organoids. A white arrow shows a liquid-filled cyst. (**B**) Live-imaging of day 14 control and 1× condition organoids showing LifeAct-GFP (green). Scale bars 200 μm. (i,ii) Enlarged captions of (i) control and (ii) 1× CHIR + BMP4 condition, showing the NE centers and the ventricle area. (**C**) LifeAct-GFP (Actin in green) and H2b-cherry (nuclei in red) and immunohistochemistry of AQP1 (blue) on day 14 control and 1× condition organoids. Scale bars 50 μm.

**Figure 4 bioengineering-07-00164-f004:**
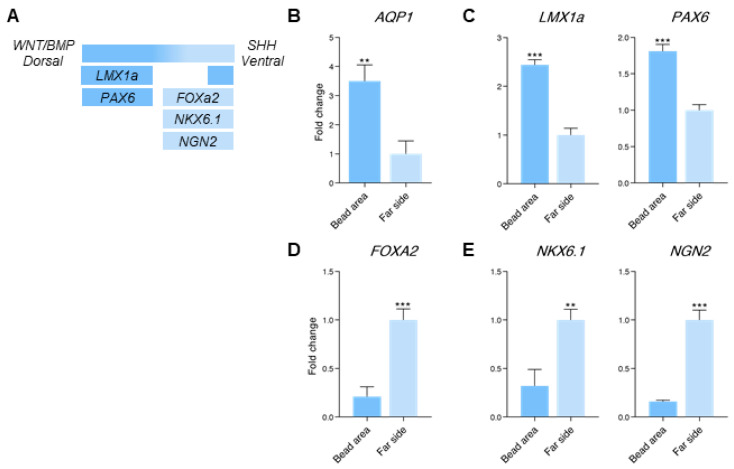
Generation of dorsal and ventral midbrain identities (2× condition). (**A**) Schematic of gene expression patterning of the dorso-ventral axis in the developing midbrain. (**B**–**E**) qPCR analysis of genes that are expressed in the bead area versus the far side of the organoid (**B**) *AQP1*; (**C**) *LMA1a* and *PAX6* genes, which are mostly expressed in the dorsal midbrain; (**D**) *FOXA2*, a ventral-side marker; (**E**) *NKX6.1* and *NGN2*, which are expressed in the ventral midbrain and are regulated by *FOXA2*. Error bars represent ± SEM. N = 24 organoids per experimental group. Comparisons were analyzed using Student’s *t*-test: n.s. non-significant *p*-value > 0.05, * *p*-value < 0.05, ** *p*-value < 0.01, *** *p*-value < 0.001.

**Figure 5 bioengineering-07-00164-f005:**
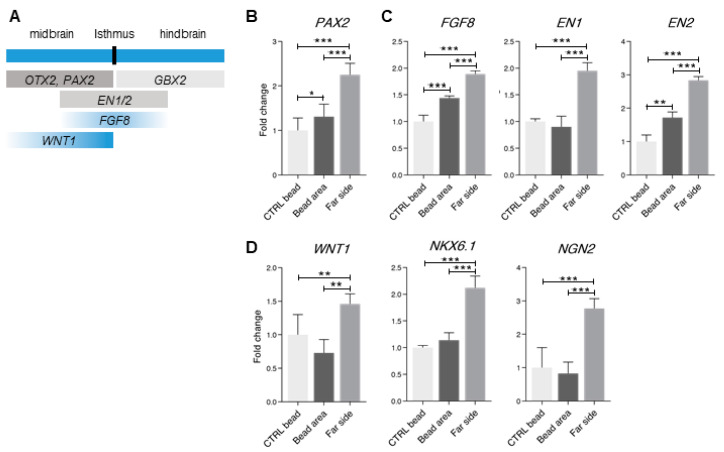
Emergence of isthmus-related genes in human brain organoids (4× condition). (**A**) Schematic of the genes which are expressed during the development from the midbrain (anterior/rostral side) through the isthmus border and to the hindbrain (posterior/caudal side). (**B**–**D**) qPCR analysis of genes which are expressed in the bead area versus the far side of the organoid and compared to the controls. (**B**) Expression of the midbrain gene *PAX2*. (**C**) Expression of the midbrain-hindbrain border genes *FGF8*, *EN1* and *EN2*. (**D**) Expression of the ventral-related genes *WNT1*, *NKX6.1* and *NGN2*. Error bars represent ± SEM. N = 24 organoids per experimental group. Comparisons were analyzed using *ANOVA* with post-hoc Tukey’s multiple comparisons test (DF = 15): n.s. non-significant *p*-value > 0.05, * *p*-value < 0.05, ** *p*-value < 0.01, *** *p*-value < 0.001.

**Figure 6 bioengineering-07-00164-f006:**
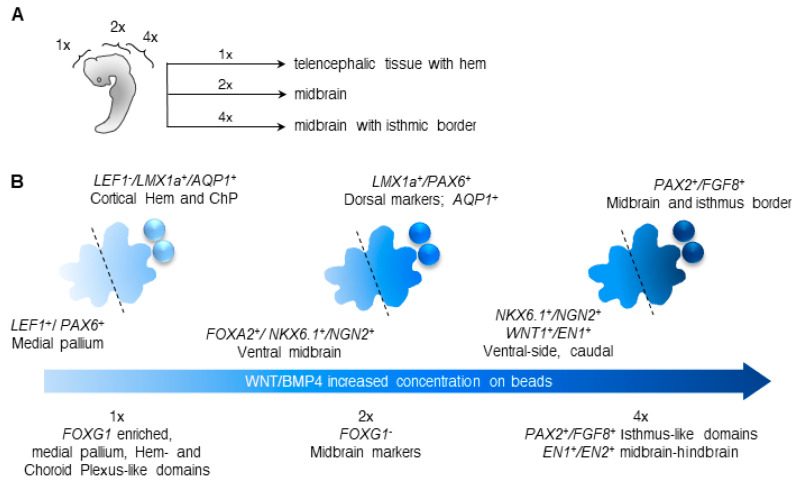
WNT/BMP4 gradients regulate the AP and the DV patterning of human brain organoids. (**A**) Schematic of an embryo, showing the different concentrations of CHIR and BMP4 which were used in the above-described experiments. The 1× condition represents a telencephalon-enriched organoid with cortical hem and ChP characteristic markers. The 2× condition generates dorsal and ventral midbrain-related fates. The 4× condition is characterized by midbrain fates with isthmic border markers. (**B**) Summary of the achieved polarization of human brain organoids. Increasing concentrations of CHIR and BMP4 has generated organoids with elevated forebrain, hem and ChP markers; dorsal and ventral midbrain genes; and isthmic (MHB border) domains. Each organoid was polarized, and contained a dorsal area (in close proximity to the beads) and ventral area (at the far side).

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
