# Peer review of "Toward Spatial Identities in Human Brain Organoids-on-Chip Induced by Morphogen-Soaked Beads"

_bioengineering, 2020, doi:10.3390/bioengineering7040164_

Round 1
Reviewer 1 Report
General comments:
The system is well described and the use of factor soaked beads to drive regional identity formation in neural organoids is of interest - However, most of the organoid mass is not accessible to live imaging and it is not clear that protein expression induced in the organoid cells matches the changes and patterning indicated by the relatively crude close and far separations for the RNA expression shown. The study would be much strengthened by including images of protein expression for the regional makers used. It is also not clear from the timelines and images from this study that the system allows the degree of tissue organization seen in other neural organoid protocols or that the bead induced signaling results in sustained changes in organoid regional identity.
Minor points
Line 19 - change to include iPSCs - use pluripotent stem cells
Line 33 remove a pool of
Line 106 add more details of the NHSM media to assist researchers trying to replicate
Line 119 add catalog number for Matrigel used and coating protocol
Line 126 add more details of neural media and supplements do not just reference previous work
Line 164 Add details of fixation protocol
Line 229 - remove on chip
Line 267 - I would like to see additional examples of the b-catenin expression to allow assessment of signaling across other organoids with multiple extensions
Line 298 Addition of immunohistochemistry to illustrate the patterns of protein expression in response to the bead induced signaling should be added to supplement the global RNA expression data - this would add to the strength of the system for initiating tissue layer formation
Fig 2 - Does the signaling actually require14 days? What happens if only 7 days of signaling are used with the same 14 day readout time or longer - initiating the formation of “endogenous” signaling centers would add to the utility of the system
Fig 4 similar to Figure 2/3 - immunohistochemistry for the markers used should be shown to add to the demonstration that the bead induced signaling does change cell phenotypes in a patterned fashion.
Supplementary
Add the expected product size and gene target for primers to help researchers repeating the experiments
Reviewer 2 Report
This paper seeks to develop more physiologically-relevant neural organoids for developmental neurobiology research by placing morphogen-soaked beads next to developing neural organoids to create neurochemical gradients of key developmental molecules (CHIR, a Wnt agonist, and BMP4).
The authors describe a platform where organoids are placed and cultured for days/weeks, with live imaging capability to monitor neural membranes and nuclei. With the addition of different concentrations of morphogen-soaked beads placed next to the organoids on the chip, the authors state that they are able to produce organoids with more telencephalic, midbrain and midbrain-hindbrain border phenotypes.
This paper nicely describes the rationale and methodology to create region-specific organoids, and results are clear. The division of organoids into 'near bead' vs. 'far side' is simplistic in terms of creating a morphogen gradient, but is nevertheless useful. I have some concerns about methodology, plus some scientific rationale and statistical details are missing. However, the science appears sound and the ability to create more localized neural regions by simple addition of microbeads could prove methodologically easy and useful for the field of neural developmental biology and those seeking to model neurodevelopmental disorders localized to specific developing brain regions.
Concerns:
- The scientific rationale for 1x, 2x and 4x concentrations of morphogens representing different developing brain regions (telencephalon, midbrain, midbrain/hindbrain) are not well-explained and need further information. What informed the 1x concentration levels? Why is this equivalent to telencephalon? Are 2x and 4x arbitrarily chosen? Some of this information is presented in Figure 6, which would benefit from being moved into the introduction.
- How close in vivo are the 'organizing centers' that secrete morphogens relative to the closeness of the morphogen-soaked bead in culture? Are the concentrations appropriate? How far do morphogens diffuse in culture, and can it be shown that the actual molecules display a gradient (rather than the indirect outcomes of morphogen presence e.g. beta-catenin presence, gene expression)?
- The organoid size increases to ~0.8mm (it says area - how was this quantified?) over 21 days. At this stage, cells internal in the organoid would have less access to oxygen diffusing in, so how can the researchers check for and quantify this? Is the organoid core becoming necrotic at this time? How would this affect organoid growth and response? No results on cell hypoxia or cell death markers are presented; this should be addressed (or it should be justified as to why it is not addressed).
- Why not include immunohistochemistry for Wnt and BMP receptors on the cells? The authors include data looking at beta-catenin but this is a downstream effector, and the immunofluorescence seen in figure 2B could be imaging artifact.
- What were the control conditions? No beads, beads + some other protein, etc? Were control cultures also subject to a bead change at day 11? Why are the controls not presented in figure 4 (dorsal vs. ventral midbrain identities, 2x condition)?
Minor issues:
- what size (+/- um) are the organoids after day 7 of culture, when they are placed on the chip?
- The immunostaining protocol is incomplete. What washes were done and when? Which primary antibodies were used, and from where? What antibody dilutions were used for secondaries? What non-specific binding control studies were done?
- The F value and degrees of freedom should be reported for each ANOVA, and post-hoc tests used to identify differences between groups stated.
Reviewer 3 Report
Towards spatial identities in human brain organoids-on-chip induced by morphogen-soaked beads.
The presented manuscript reports the design and fabrication of a human brain organoid-on-chip induced by morphogen-soaked beads. For that, the authors described the construction of a novel on-chip platform of brain models, applying increasing concentrations of CHIR (WNT agonist) and BMP4 on the beads that resulted in the emergence of dorsoventral and rostrocaudal/anteroposterior identities.
The authors present a well-structured manuscript and relevant experimental work with interesting findings. Showing that the presented platform can be a first step for the development of representative brain-on-a-chip devices to study developmental concepts and disease modeling.
Overall, the manuscript is an interesting contribution for the readers of Bioengineering and should be considered for publication as presented.
Round 2
Reviewer 1 Report
Thank you for responding to the comments on the original submission - I will recommend acceptance of the revised manuscript